# Stem Cell Bioengineering with Bioportides: Inhibition of Planarian Head Regeneration with Peptide Mimetics of Eyes Absent Proteins

**DOI:** 10.3390/pharmaceutics15082018

**Published:** 2023-07-26

**Authors:** Sarah Jones, Bárbara Matos, Sarah Dennison, Margarida Fardilha, John Howl

**Affiliations:** 1Research Institute in Healthcare Science, Faculty of Science & Engineering, University of Wolverhampton, Wulfruna Street, Wolverhampton WV1 1LY, UK; john_871@hotmail.co.uk; 2Laboratory of Signal Transduction, Department of Medical Sciences, Institute of Biomedicine—iBiMED, University of Aveiro, 3810-193 Aveiro, Portugal; barbaracostamatos@ua.pt (B.M.); mfardilha@ua.pt (M.F.); 3School of Pharmacy and Biomedical Sciences, University of Central Lancashire, Preston PR1 2HE, UK; srdennison1@uclan.ac.uk

**Keywords:** cell penetrating peptide, bioportide, eyes absent protein, neoblast, planarian, confocal microscopy, stem cell

## Abstract

Djeya1 (RKLAFRYRRIKELYNSYR) is a very effective cell penetrating peptide (CPP) that mimics the α5 helix of the highly conserved Eya domain (ED) of eyes absent (Eya) proteins. The objective of this study was to bioengineer analogues of Djeya1 that, following effective translocation into planarian tissues, would reduce the ability of neoblasts (totipotent stem cells) and their progeny to regenerate the anterior pole in decapitated *S. mediterranea*. As a strategy to increase the propensity for helix formation, molecular bioengineering of Djeya1 was achieved by the mono-substitution of the helicogenic aminoisobutyric acid (Aib) at three species-variable sites: 10, 13, and 16. CD analyses indicated that Djeya1 is highly helical, and that Aib-substitution had subtle influences upon the secondary structures of bioengineered analogues. Aib-substituted Djeya1 analogues are highly efficient CPPs, devoid of influence upon cell viability or proliferation. All three peptides increase the migration of PC-3 cells, a prostate cancer line that expresses high concentrations of Eya. Two peptides, [Aib^13^]Djeya1 and [Aib^16^]Djeya1, are bioportides which delay planarian head regeneration. As neoblasts are the only cell population capable of division in planaria, these data indicate that bioportide technologies could be utilised to directly manipulate other stem cells in situ, thus negating any requirement for genetic manipulation.

## 1. Introduction

Cellular and tissue permeability barriers are significant caveats for bioengineering strategies to develop bioactive agents and exploit novel intracellular drug modalities. As recently reviewed [1,2,3], cell penetrating peptides (CPPs), often polycationic linear sequences of 12–24 amino acids, are a versatile technology that can overcome the common biophysical constraint of ineffective intracellular access. Bioportides, CPPs with intrinsic bioactivities and so partially distinct from conventional inert vectors, accrete within eukaryotic cells to influence protein function and impact cell biology [4]. The molecular organisation of bioportides commonly includes mimetic sequences derived from functional protein domains to serve as selective modulators of intracellular protein-protein interactions (PPIs; [4,5,6]). Hence, bioportide technologies, presumably acting by a dominant-negative mechanism, enable both the understanding and discrete manipulation of intracellular signalling pathways that regulate cellular biology [5,6].

The triploblastic bilateral planarian *Schmidtea mediterranea* is a very common model organism employed to address fundamental cellular processes that include tissue regeneration and the differentiation of neoblasts, relatively small pluripotent stem cells of mesenchymal origin [7,8]. Indeed, a total of 20–30% of *S. mediterranea* cells are neoblasts, the only planarian cell type capable of mitotic division [7,8]. A comprehensive *S. mediterranea* genomic database (SmedGD 2.0) confirmed that planaria are genetically more like vertebrates than both *Drosophila melanogaster* and *Caenorhabditis elegans* [9,10]. *S. mediterranea* is also a rigorous three-dimensional model to analyse the import of CPPs and bioportides into complex tissues presenting both physical and metabolic barriers [11].

In common with studies of arginine-rich peptides derived from both RNA- and DNA-binding proteins [12], planarian proteins that collectively control head remodelling and eye regeneration following decapitation are a viable source of cationic CPP vectors [11]. Djeya1 (RKLAFRYRRIKELYNSYR), an octadecapeptide sequence mimicking a highly conserved domain of eyes absent (Eya) proteins, is a particularly efficient and seemingly inert example of such a CPP vector sequence. Three days after head amputation, fluorescent Djeya1 effectively enters the unpigmented *S. mediterranea* blastema, a transient and heterogenous cell mass responsible for head morphogenesis, to penetrate deeper along the dorsal ventral axis [11]. Thus, CPPs such as Djeya1 provide the means to target bioactive agents to differentiate post-mitotic neoblast progeny [13] in addition to epithelial precursor cells or neoblast-derived mesenchymal cells.

Djeya1 mimics part of the α5 helix within the evolutionary conserved C-terminal Eya Domain (ED) of Eya proteins [14,15]. The ED domain acts as a transcriptional regulator known to bind proteins such as Dachshund and Sine Oculus [14,15]. Hence, the major objective of this study was to determine whether the bioengineering of Djeya1 analogues could provide rhegnylogically organised bioportides, CPPs in which the pharmacophores that enable cellular penetration and those essential for bioactivity are discontinuously organised. This is in contrast to a sychnologic organisation in which the pharmacophores for penetration and bioactivity are distinct. Moreover, given that Arg is the quantitatively dominant amino acid at PPI interfaces, there will clearly be some pharmacophores which are both cell penetrant and bioactive within a rhegnylogic organisation [4,5]. We hypothesized that analogues of Djeya1 with enhanced helicity would more effectively mimic ED to modulate PPIs and inhibit anterior pole and eye regeneration in *S. mediterranea* [11,14,15,16]. To promote helicity, we introduced the helicogenic amino acid α-aminoisobutyric acid (Aib) at three sites within Djeya1 (I^10^, L^13^, S^16^) where there is species-specific heterogeneity of the protein sequence in Eya proteins. All three bioengineered Djeya1 analogues are highly efficient CPPs. Two of these, [Aib^13^]Djeya1 and [Aib^16^]Djeya1, inhibited planarian head regeneration whilst [Aib^10^]Djeya1 was inert. As neoblast populations are the only cell type capable of division in planarians [7,8,13,16,17,18], our data indicate that both [Aib^13^]Djeya1 and [Aib^16^]Djeya1 directly influence the biology of stem cells. Furthermore, cellular assays confirmed that this action was unlikely the consequence of a detrimental influence of bioportides upon cellular viability, proliferation, or motility. Hence, the same bioportides might modulate the morphogenesis of mammalian stem cells regulated by eyes absent proteins [19].

## 2. Materials and Methods

### 2.1. Peptide Synthesis and Secondary Structure Analysis

#### 2.1.1. Microwave Enhanced Solid Phase Peptide Synthesis

Aib-substituted Djeya1 analogues (Figure 1) were synthesized using a CEM Liberty Blue Synthesizer, equipped with a UV analyser (CEM Microwave Technology Ltd., Buckingham, UK) in dimethylformamide (DMF; Cambridge Reagents, Barton upon Humber, UK) on Rink-amide 4-methylbenzhydrylamine (MBHA) resin (Novabiochem, Beeston, UK) to generate peptide amides [11,20]. N,N′-diisopropylcarbodiimide (DIC; Sigma-Aldrich, Gillingham, UK) was used as a condensation reagent with the additive ethyl 2-cyano-2-(hydroxyimino)acetate (Oxyma; [21]; Novabiochem, Beeston, UK) plus a trace concentration of N,N-diisopropylyethylamine (DIPEA; Sigma-Aldrich, Gillingham, UK). Syntheses on a 0.1 mmole scale were performed using a 5-fold molar excess of AA/DIC/Oxyma with 0.1 molar equivalent of DIPEA in a final volume of 4 mL. Single coupling of most Fmoc-AAs (Novabiochem, Beeston, UK) was achieved at 90 °C/120 s. To overcome any steric hindrance, both the dialkylated Aib and following AA were double coupled using an identical protocol. Arg was routinely double coupled at 75 °C/300 s. A standard deprotection cycle with 20% (*v*/*v*) Piperidine (Sigma-Aldrich, Gillingham, UK) in DMF was 90 °C/60 s routinely monitored by determining the UV absorption at 301 nm of dibenzofulvene-piperidine adducts [20]. Amino-terminal acylation of Aib-substituted Djeya1 analogues with 6-carboxytetramethylrhodamine (Novabiochem, Beeston, UK) yielded fluorescent analogues with excellent photo-stability and pH-insensitive fluorescence. All peptides described herein were purified to apparent homogeneity by reverse phase HPLC [4,6,22].

#### 2.1.2. Circular Dichroism

Secondary structure analyses of Djeya1 peptides were performed on a Jasco J-815 CD spectrometer (Jasco Ltd., Heckmondwike, UK). Peptides (0.01 mg mL^−^^1^) were dissolved in either 1 × phosphate buffered saline (PBS) at pH 7.4 or in a 2,2,2-tetrafluoroethanol (TFE) and 1 × PBS mixture (50.0% *v*/*v*). All buffers were prepared using ultra-pure water (resistivity 18 MΩ cm), PBS and TFE (Scientific Laboratory Supplies Ltd., Nottingham, UK). CD experiments were also performed at peptide:lipid ratios of 1:100, whilst 1,2-dimyristoyl-sn-glycero-3-phosphatidylcholine (DMPC) and 1,2-dimyristoyl-sn-glycero-3-phosphatidylserine (DMPS) (5 mg mL^−^^1^) (Avanti Polar Lipids, Alabaster, AL, USA) were dissolved separately in chloroform (HPLC grade, VWR International Ltd Lutterworth, UK) and dried under N_2_ gas. The lipid film was rehydrated using 1 × PBS, pH 7.5 for 1 h or until the solution was no longer turbid. The solution then underwent 5 cycles of freeze thaw before the lipid/peptide samples were prepared by the addition of peptide stock solution (final concentration 0.01 mg mL^−^^1^). Far UV CD spectra (180 nm to 260 nm) were recorded at 20 °C using 0.5 nm intervals, a bandwidth of 1 nm, a scan speed of 50 nm min^−1^ and a 10 mm path-length cell. Ten accumulations were selected to average a single spectrum. Samples without peptide were prepared for background spectral subtraction. Data, baseline-subtracted and averaged CD, were analyzed using the Dichroweb server CDSSTR and SELCON3 reference dataset 3 [23,24,25]. To generate percentage structure averages, these analyses were repeated four times.

### 2.2. Schmidtea mediterranea Culture and Head Morphogenesis

#### 2.2.1. Planarian Maintenance

*S. mediterranea*, hermaphroditic sexual strain, were a kind gift from Kerstin Bartscherer (Max Planck Institute for Molecular Medicine, Münster, Germany). As previously described [11], animals were maintained at 20 °C in planarian artificial medium (PAM, [26]) containing NaCl (1.6 mM), MgSO_4_ (1 mM), MgCl_2_ (0.1 mM), KCl (0.1 mM), NaHCO_3_ (1.2 mM), and CaCl_2_ (1 mM) in ultrapure water with gentamycin (3 µg mL^−^^1^). For maintenance, animals were routinely fed finely minced calf’s liver at two-weekly intervals.

#### 2.2.2. Anterior Pole Regeneration and Eye Development Assay

Adult planaria, 4–6 mm in length, and selected for experimentation were starved 5 days prior to amputation. Transverse amputation of planarian heads, post-auricle and pre-pharynx, induced blastema formation in the trunk section, leading to eye regeneration and head remodeling [11].

Post amputation, planaria were immediately treated with exogenously added Djeya1 analogues in 35 mm diameter × 10 mm depth culture dishes (VWR International Ltd., Lutterworth, UK) to a final volume of 4 mL PAM and maintained at 20 °C. PAM, containing Djeya1 analogues, was replenished 4 days post amputation. Animals were allowed to regenerate for 7–8 days post amputation, whilst observations were performed daily to assess any morphological variations. To document the influence of bioportides upon head morphogenesis, living specimens were observed with a Swift SM series stereo microscope equipped with Moticam BTU assembly for image acquisition (Swift Microscope World, Carlsbad, CA, USA).

### 2.3. Biological Characterization of Aib-Substituted Djeya1 Analogues

In the absence of a transformed neoblast cell line or a readily isolated primary culture, U373MG astrocytic tumour cells were employed as a robust model to determine CPP internalization and discount any unwanted cytotoxic effects [20,27]. Transformed cell lines known to overexpress Eya proteins were used to quantify the impact of Djeya1 analogues on cell migration (PC-3 prostate cancer cells [28]) and proliferation (U251 glioblastoma cells [29,30]).

#### 2.3.1. Cell Culture Maintenance

U373MG and U251 cells were both maintained in Dulbecco’s Modified Eagle’s Medium (DMEM) containing L-glutamine (0.1 mg mL^−^^1^) (Sigma-Aldrich, Gillingham, UK) and PC-3 in Roswell Park Memorial Institute media-1640 (RPMI-1640) with L-glutamine (Gibco, Life Technologies, Carlsbad, CA, USA). Both media were supplemented with foetal bovine serum (FBS) 10% (*v*/*v*), penicillin (100 U mL^−^^1^), and streptomycin (100 µg mL^−^^1^), and all cell lines were maintained in a humidified atmosphere of 5% CO_2_ at 37 °C.

#### 2.3.2. Qualitative Peptide Uptake Analyses

Live confocal cell imaging analyses were employed to avoid fixation artefacts and establish the intracellular distributions of tetramethylrhodamine (TAMRA)-conjugated Djeya1 analogues [22]. U373MG cells were maintained as above, transferred to 35 mm sterile glass base dishes (NuncTM, Fisher Scientific, Loughborough, UK), and grown to ∼75% confluence. Cells were washed in phenol red-free DMEM prior to treatment with (TAMRA)-conjugated Djeya1 analogues diluted to a final concentration of 5 μM in phenol red free media. Treated cells were maintained at 37 °C in a humidified atmosphere of 5% CO_2_ for the designated time periods. For the 1 h incubations, cells were also treated with CellMaskTM (5 µg mL^−^^1^; Molecular Probes, Thermo Fisher Scientific, Waltham, MA, USA) for an additional 5 min prior to observation using a Zeiss LSM 880 microscope equipped with live cell imaging chamber (Zeiss, Cambridge, UK). A total of 5 h incubations were also observed using a photomultiplier for transmitted light (T-PMT).

#### 2.3.3. Quantitative Peptide Uptake Analyses

U373MG and U251 cells were maintained as above and then transferred to 6-well plates and grown to 80% confluence. Cells were washed and maintained in phenol red-free DMEM and subsequently treated with TAMRA-conjugated Djeya1 analogues at final concentrations of 1 μM, 2.5 μM, and 5 μM for 1 h in culture conditions as above. Cells were washed four times, detached with 300 μL of 1% (*w*/*v*) trypsin (without phenol red) at 37 °C, collected by centrifugation, and lysed in 300 μL 0.1 M NaOH for 2 h on ice. A total of 250 μL of each sample cell lysate was transferred to a black 96-well plate and analysed using a Thermo Fischer Scientific (Loughborough, UK) Fluoroskan Ascent FL fluorescence spectrophotometer (λAbs 544 nm/λEm 590 nm).

#### 2.3.4. Cytotoxicity Assays

When employing U373MG cells, cytotoxicity was quantitatively assessed using the 3-(4,5-dimethylthazol-2-yl)-2,5-diphenyl tetrazolium bromide (MTT; Sigma-Aldrich, Gillingham, UK) conversion assay [31,32]. U373MG cells were cultured as above in 96-well plates and treated with peptides (0.1–30 μM) in DMEM without FBS. At 4 h post treatment, cells were incubated for a further 3 h with MTT (0.5 mg mL^−^^1^). The insoluble formazan product was solubilised with DMSO (Sigma-Aldrich, Gillingham, UK) and MTT conversion determined by colorimetric analysis at 540 nm (Labsystems Multiskan Ascent 354 Microplate Reader, Thermo Fischer Scientific, Loughborough, UK). Cellular viability was expressed as a percentage of those cells treated with vehicle (medium) alone.

PC-3 prostate cancer cells were seeded in supplemented RPMI-1640 (as described above) into 96-well plates (1.0 × 10^4^ cells/well) and maintained in a humidified atmosphere of 5% CO_2_ at 37 °C for 24 h. Thereafter, cells were treated with 3 μM, 10 μM, and 25 μM of Djeya1 analogues in RPMI-1640 medium without FBS for 24 h, whilst maintaining culture conditions. Untreated (medium alone) cells were used as a control representing 100% viability. For the final hour of incubation, 10 µL of PrestoBlue^TM^ Cell Viability reagent, resazurin-based reagent (Thermo Fisher Scientific, Waltham, MA, USA) was added to each well. A total of 100 µL of culture medium from each well was transferred to a black bottomed 96-well plate and the fluorescence at λ_abs_ = 560 nm and λ_em_ = 590 nm was measured using a microplate reader (Tecan Infinite^®^ 200 PROseries, Mannedorf, Switzerland). Cellular viability was expressed as a percentage of those cells treated with vehicle (medium) alone. Three independent experiments with five replicates for each condition were performed.

#### 2.3.5. Cellular Proliferation

Cellular proliferation was determined using measurements of cell viability and employed the MTT conversion assay as described above. U251 cells were cultured as previously described, grown to anastomose in 96-well plates, and treated from 4–72 h with Djeya1 analogues (3 μM, 10 μM, 25 μM) in DMEM supplemented with FBS 10% (*v*/*v*) and maintained in a humidified atmosphere of 5% CO_2_ at 37 °C. Cell viability was expressed as absorbance minus background at 540 nm. Cells treated with medium alone were also included at each designated time point and acted as a comparator when constructing growth curves.

#### 2.3.6. Cell Migration Assays

PC-3 cells (1.00 × 10^5^) were seeded in supplemented RPMI-1640 (as described above) in 24-well plates and incubated for 24 h at 37 °C in a humidified atmosphere of 5% CO_2_. Confluent cells monolayers were wounded by scratching lines with a 200 µL pipette tip. Cells were washed in phosphate-buffered saline (PBS) and incubated with fresh medium without FBS containing 3 µM or 25 µM of Djeya1 analogues for 48 h. Untreated cells were also included in the assay. Photographs were taken under ×40 magnification for EVOS^TM^ M5000 imaging (Thermo Fisher Scientific, Waltham, MA, USA), immediately after wound incision and after 48 h. Results were expressed as a percentage of wound closure, relative to the time at 0 h. Three independent replicates of each condition were performed.

### 2.4. Graphical Representations and Statistical Analyses

These were performed using GraphPad Prism 9 software. For cytotoxicity and cellular proliferation assays, statistical analyses of changes in cellular viability employed an unpaired, 2-tailed, non-parametric Mann–Whitney test. Statistical analyses of changes in wound closure used a paired, 2-tailed, non-parametric Wilcoxon matched pairs signed rank test.

## 3. Results

### 3.1. Site-Directed Bioengineering of the Djeya1 CPP

The entire ED of Eya proteins is exceptionally well conserved to provide a unique tyrosine phosphatase activity within a multi-domain protein classified as a haloacid dehalogenase [13,14,15,33]. The transcriptional activity of Eya proteins, dependent upon PPIs notably with Dachshund and SIX family members, appears to be regulated by the dephosphorylation activity of the ED [14,15,33,34]. The CPP Djeya1 (Table 1; [11]), identical in planarians *Dugesia juponica* and *S. mediterranea*, mimics helix 5 of the ED conserved within plants, fungi, and animals, a region that could participate in PPIs required for trancriptional activity [33,34]. Within a consensus sequence of this helical domain, three variable positions (10,13,16) were selected for Aib-substitution as a bioengineering strategy to promote helicity. Very conservative changes are indicated at positions 10 (I/V) and 13 (L/I) with more chemical variability at site 16 (S/T/A/Q). We did not introduce Aib at position 18 (R/K) as the loss of cationic charge could impact on the uptake efficacy of the modified octadecapeptide. The sequences of these peptides and others used in this study are presented in Table 2.

The synthesis and purification of Djeya1, Tat (denotes the Tat peptide, Tat ^48−60^), C105Y, and mitoparan have been described elsewhere [11]. All peptides were synthesized as C-terminal amides. The identities of three Aib-substituted Djeya1 analogues were confirmed by mass spectrometry, Agilent 6200 TOF: [Aib^10^]Djeya1, calculated 2403.8 Da, observed 2403.4 Da; [Aib^13^]Djeya1, calculated 2403.8 Da, observed 2403.4 Da; and [Aib^16^]Djeya1, calculated 2429.9 Da, observed 2429.4 Da. Spectra are presented in Appendix A.

### 3.2. CD Spectral Analyses of Peptide α-Helicity

CD conformational analyses indicated that in aqueous solution (PBS, pH 7.4), Djeya1, [Aib^10^]Djeya1, [Aib^13^]Djeya1, and [Aib^16^]Djeya1 displayed spectra typical of α-helical and random coil structure (Figure 1a). Figure 1a shows two minima at 220 nm and 207 nm, which is typical of α-helical peptides, and a negative band at 195 nm, which is characteristic of a random coil structure. The low intensities of these bands indicate a mixture of these structures, which is confirmed by further analysis in Table 3a. The estimated α-helical content of these peptides was 43.00 ± 1.73% for Djeya1, 41.33 ± 1.15% for [Aib^10^]Djeya1, 40.67 ± 1.52% for [Aib^13^]Djeya1, and 39.67 ± 4.04% for [Aib^16^]Djeya1.

A key determinant in the membrane interaction of these peptides likely involves the adoption of secondary structures in the anisotropic environment of the interface [35]. This phenomenon is often investigated using TFE, a membrane-mimicking solvent with an α-helix-enhancing effect [36,37]. Figure 1b indicates that in a 50% (*v*/*v*) TFE/PBS (pH 7.4) mixture, all Djeya1 analogues adopted conformations characterized by two minima near 205 and 225 nm, respectively, and a maximum at 193 nm, which is typical of α-helical peptides [38]. Further analysis of these CD spectra showed that enhanced α-helicity was observed for Djeya1, [Aib^10^]Djeya1, and [Aib^16^]Djeya1 (45.50 ± 4.94%, 51.50 ± 0.07 and 46.00 ± 1.41%; Table 3b) with the remaining structural contributions to the peptide provided by random coil and β-type architectures. However, the presence of 50% TFE/PBS (*v*/*v*) mixture did not enhance the α-helical structure of [Aib^13^]Djeya1.

**Figure 1 pharmaceutics-15-02018-f001:**
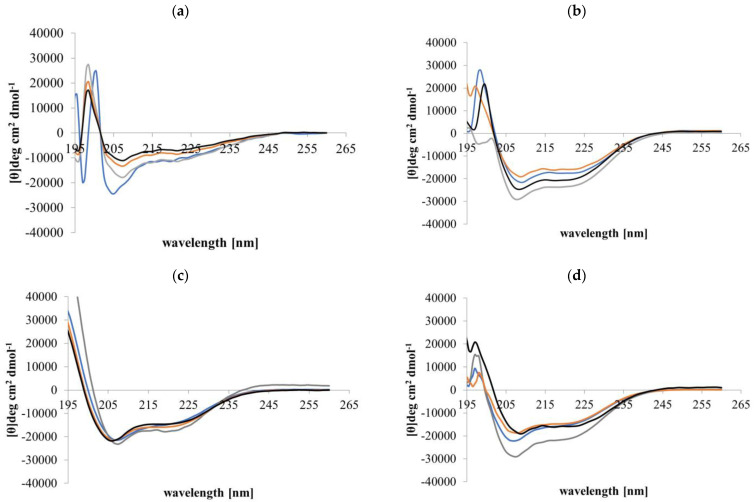
Conformational behavior of Djeya1 analogues. (**a**) Spectra obtained in PBS. (**b**) Spectra obtained in 50% (*v*/*v*) TFE. (**c**) Spectra obtained in the presence of DMPC vesicles. (**d**) Spectra obtained in the presence of DMPS vesicles. Djeya1, blue; [Aib^10^]Djeya1, grey; [Aib^13^]Djeya1, orange; [Aib^16^]Djeya1, black.

It is widely accepted that peptides, including CPPs, adopt and enhance an α-helical structure in the presence of a lipid. Therefore, the structure of the peptides used here (Table 3c,d) were analysed in the presence of lipid vesicles at a lipid:peptide ratio of 100 to 1. Figure 1c shows that in the presence of the zwitterionic lipid DMPC, peptides underwent conformational change and adopted an α-helical structure, which ranged from 42% to 45% (Table 3c). In the presence of the anionic lipid DMPS, Figure 1d shows that Djeya1 and [Aib^10^]Djeya1 displayed 47% and 63% α-helicity, respectively (Table 3d), indicating that the presence of an anionic lipid can strongly initiate α-helix formation by these peptides at the lipid interface. No difference in percentage α-helicity was observed for [Aib^13^]Djeya1 and [Aib^16^]Djeya1 in the presence of these membranes.

### 3.3. Exogenous Application of [Aib^13^]Djeya1 and [Aib^16^]Djeya1 Delays Anterior Pole Regeneration and Eye Development in S. mediterranea

Immediately following transverse amputation (Figure 2a), planaria were treated with 25 μM of Aib-substituted Djeya1 analogues and monitored for morphological changes over an 8-day period. Additional treatments at the same concentration were administered on day 4 to compensate for any premature proteolytic degradation of the peptides. On day 8, planaria treated with [Aib^13^]Djeya1 and [Aib^16^]Djeya1 were unresponsive to light and regeneration of the anterior pole, including the eye spots, containing photoreceptors and the pigment cup, were absent (Figure 2c,d), compared to control planaria treated with PAM alone (Figure 2b). On day 9, 67% (*n* = 9) of both the [Aib^13^]Djeya1 and [Aib^16^]Djeya1-treated animals had commenced the development of photoreceptors within the unpigmented blastema. A total of 30 days post amputation, in PAM, all of the [Aib^16^]Djeya1-treated planaria had fully regenerated, though only 67% (*n* = 9) of the [Aib^13^]Djeya1-treated had achieved full regeneration and 33% remained unregenerated and in stasis. Further observations included an absence of generalized necrosis without excessive mucous production as a general indicator of animal stress. In these assays both Djeya1 and [Aib^10^]Djeya1 were inactive.

### 3.4. Aib-Substituted Djeya1 Analogues Are Efficient CPPs

Red fluorescent TAMRA-conjugated peptides were utilised to compare the efficacy of cellular uptake. Comparative investigations utilised Tat and C105Y CPPs [11] as positive controls.

#### 3.4.1. Qualitative Peptide Uptake Analyses

Confocal analyses compared the intracellular distribution of peptides in U373MG cells [11]. Observations after 1 h of exogenous peptide application (Figure 3a) revealed a similar distribution of Djeya1 analogues, which were, in part, perinuclear, and which were enhanced compared with both Tat and C105Y controls. Even after an extended period of 5 h (Figure 3b), Djeya1 analogues were absent from the cell nucleus.

#### 3.4.2. Quantitative Peptide Uptake Analyses

These studies (Figure 4) compared the intracellular uptake of Aib-substituted Djeya1 analogues into U373MG cells and U251 cells, the latter of which is reported to overexpress Eya proteins. All three Aib-substituted Djeya1 analogues are highly efficient CPPs, broadly comparable with Djeya1 and C105Y but superior to Tat, though they displayed a different rank order of uptake efficacy in the two cell lines: U373MG [Aib^16^]Djeya1 = [Aib^10^]Djeya1 > [Aib^13^]Djeya1; U251 [Aib^13^]Djeya1 > [Aib^16^]Djeya1 > [Aib^10^]Djeya1. Of particular note is the increased uptake efficacy of [Aib^13^]Djeya1 into U251cells (Figure 4b) compared with U373MG cells (Figure 4a).

### 3.5. Cytotoxicity of Djeya1 Analogues

A negative influence of peptides upon the viability of neoblasts could underlie their ability to inhibit head morphogenesis in *S. mediterranea* (Figure 2). As indicated in Figure 5, Djeya1 analogues had no influence upon the viability of U373MG cells following exposure to peptides at concentrations of 0.1–30 µM for 4 h. Treatment of PC-3 cells with Djeya1 analogues for a longer time of 24 h similarly induced no significant changes in cellular viability (Appendix A).

### 3.6. Impact of Djeya1 Analogues upon U251 Cellular Proliferation

Planarian neoblasts replenish lost organs and tissues by proliferation within hours of any injury [7,8,39,40]. Many wound-induced transcriptional changes necessary for these homeostatic events are regulated by the activity of extracellular signal-regulated kinase (ERK) [29,40]. U251 has been reported to have a high endogenous expression of Eya2, and siRNA knockdown of this protein decreases the proliferation and invasion of these cells, whilst Eya2 positively regulates p42/44 MAPK activity [29]. We have previously reported [11] that p42/44 inhibition prevents anterior pole regeneration in *S. mediterranea*. Thus, in these experiments, we compared the impact of Aib-substituted Djeya1 analogues upon the proliferation of U251 cells employing the p42/44 MAPK inhibitor U1026 as a positive control. As revealed in Figure 6, bioengineered Djeya1 analogues had minimal impact upon cellular proliferation and, whilst statistically significant data were recorded at some higher peptide concentrations, this influence was independent of concentration.

### 3.7. Aib-Substituted Djeya1 Analogues Enhance Cell Migration

Stem cell migration is considered a fundamental process of tissue maintenance in metazoans and of particular importance following tissue loss and wounding. Migrating neoblasts and their progeny adopt distinctive behaviours to selectively regenerate appropriate missing tissues in planaria and form the regeneration blastema where differentiation is completed [40,41]. Thus, we hypothesized that bioportides influencing head morphogenesis in *S. mediterranea* might adversely inhibit cellular migration in PC-3 cells that express high levels of Eya. Contrary to these expectations, all Aib-substituted Djeya analogues enhanced cellular migration, measured in wound closure assays, at concentrations of both 3 µM and 25 µM (Figure 7). Further details of these assays are provided in Appendix A.

## 4. Discussion

The molecular bioengineering of CPPs and bioportides can enhance intracellular uptake and confer or increase biological activity [4,5,6,42]. As recently reviewed [43], biophysical methodologies, including circular dichroism, have impacted the understanding of the structural determinants of CPP trafficking and the molecular mechanisms of bioportides. Numerous studies of the secondary structures of CPPs have highlighted the possibility that a helical conformation, perhaps induced by contact with phospholipid membranes, may support the passage of CPPs into cells [43,44]. One or more cationic alpha helices, particularly those containing statistically enriched Arg residues, are commonly located at protein-protein interaction (PPI) sites. Cation-π interactions, which stabilize these PPIs [45], commonly involve interactions between arginine and tyrosine residues. Hence, we hypothesized that Aib-substitution might induce helicity within Djeya1 to further increase penetration efficacy [11]. We also anticipated that the structural constraints imposed by Aib-substitution would enable bioengineered Djeya1 analogues, by a dominant-negative mechanism, to modulate PPIs within the ED domain. Interference with EYA functions as a transcriptional regulator would likely manifest as inefficient head morphogenesis in decapitated *S. mediterranea* [15,33,34].

This study identified two bioportides, [Aib^13^]Djeya1 and [Aib^16^]Djeya1, which delay head regeneration in the *S. mediterranea* model; [Aib^10^]Djeya1, in common with Djeya1 [11], is an effective CPP, but unable to influence planarian morphogenesis. Djeya1, in common with Aib-substituted analogues, adopts significant α-helical structure in aqueous solution that is only marginally influenced by a change to 50% (*v*/*v*) TFE and in the presence of a lipid environment. It is noteworthy that the mitochondriotoxic bioportide mitoparan, a peptide that accretes within mitochondria to promote intrinsic apoptosis, includes Aib to replace Ala at position-10 of mastoparan, a known α-helix adopting peptide [32]. Thus, the molecular bioengineering of CPPs by Aib-substitution can produce significant variations in bioactivities that are not necessarily the result of gross changes in peptide secondary structure.

The bioactivities of [Aib^13^]Djeya1 and [Aib^16^]Djeya1 cannot be readily explained by increased uptake into cells. Both qualitative and quantitative analyses show that the uptake of all Aib-substituted Djeya1 analogues are generally comparable to that of the parent CPP and better than other CPPs, including Tat and C105Y. It is, however, notable that a different rank order of penetrative efficacy was evident between U373MG and U251 cell lines, which in part may be attributable to the U251 cell line overexpressing the EYA protein [29]. We have previously proposed [22] that the propensity for cellular penetration and intracellular accumulation of CPPs involves two distinct processes: the first is translocation across the plasma membrane (hydrophobicity and cationic charge are significant factors), and the second is accretion at intracellular loci. Thus, the increased uptake of [Aib^13^]Djeya1 within U251 cells could reflect, in part, enhanced accretion at intracellular loci, particularly since [Aib^10^] Djeya1, also with a conservative aliphatic side chain substitution for Aib, showed a reduced uptake efficacy compared to [Aib^13^]Djeya1.

It would be intriguing to establish the mechanism of uptake for these novel Aib-substituted Djeya1 analogues. Following live confocal cell imaging, all analogues showed a clear punctate intracellular distribution––data which suggest an endocytotic mechanism of intracellular uptake. Further investigations, including the co-labelling with specific markers of endocytosis or micropinocytosis, in addition to quantitative uptake analyses performed at both 4 °C and 37 °C, could define a mechanism of cellular uptake. Additionally, all analogues demonstrated a distinct absence of cytotoxicity even at a peptide concentration of 30 μM. Thus, these peptides do not induce gross perturbations of the plasma membrane to induce Ca^2+^-dependent necrosis.

Efforts to define a molecular mechanism of action of both [Aib^13^]Djeya1 and [Aib^16^]Djeya are hampered both by a lack of accessible planarian neoblast cultures and knowledge of a defined function for the α5 helix of ED that they mimic. However, we are confident that our investigations can exclude peptide-induced neoblast death as a biological explanation for the impact of [Aib^13^]Djeya1 and [Aib^16^]Djeya1 on anterior pole remodeling. Neither would it seem likely that these bioportides adversely influence cellular proliferation, a process fundamental to this regenerative process [16,17,18,39,40]. The observation that all three Aib-substituted Djeya1 analogues moderately enhance cell migration (wound healing) is intriguing and worthy of further investigation. However, since [Aib^10^]Djeya1 does not inhibit heard morphogenesis in *S. mediterranea*, it would appear unlikely that this positive influence upon cell migration underlies the mechanism of action of [Aib^13^]Djeya1 and [Aib^16^]Djeya1, bioportides that delay head regeneration.

In common with the dominant negative action of many other bioportides [4,5,6,42], both sychnologic and rhegnylogic in molecular organization, we propose that [Aib^13^]Djeya1 and [Aib^16^]Djeya1 modulate head regeneration by interfering with PPIs mediated by the ED of Eya proteins. Support for this hypothesis is provided by a study of the G393S mutation of ED, identified from a patient with cataracts and both renal and optic abnormalities [46]. In the human ED, this missense mutation site is located close to R^399^, the first residue of ED mimetic Djeya1 analogues. Further analyses of this mutation indicated this site to be critical for the interaction of ED with unknown proteins that bridge well characterized PPIs between Eya and SIX family proteins [46].

Considering that there are so many similarities between planarian proteins and those expressed in higher vertebrates [47], we anticipate that bioportides able to influence the distribution and function of Eya proteins may prove valuable for other studies of physiology and pathology. For example, mutations in the human *EYA1* gene cause branchio-oto-renal (BOR) syndrome [46]. The multifunctional nature of Eya proteins also influences tumour progression through multiple mechanisms [48]. Moreover, it is probable that [Aib^13^]Djeya1 and [Aib^16^]Djeya1 could be utilized to further understand and directly influence human stem cells, thus negating the requirement for genetic manipulation.

## 5. Conclusions

With the aim of modulating planarian stem cell biology through targeting PPIs integral to anterior pole and eye regeneration, rational design of helicogenic peptides, corresponding to the α5 helix of the highly conserved ED domain, generated the bioportides [Aib^13^]Djeya1 and [Aib^16^]Djeya1. Thus far, manipulations of planarian stem cell biology have predominantly utilized siRNA interference. To the best of our knowledge, this is the first instance in which bioportide technologies have given a functional response in this three-dimensional model, a response which is unlikely to be a detrimental consequence of cytotoxicity, inhibition of cellular proliferation, or migration.

## Figures and Tables

**Figure 2 pharmaceutics-15-02018-f002:**
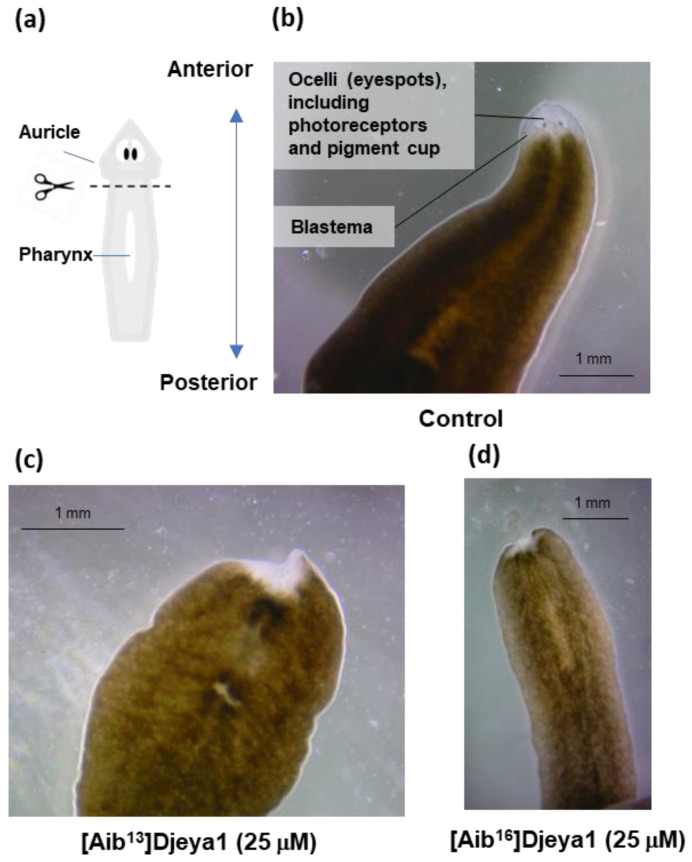
Influence of bioportides upon head remodelling. Transverse amputation at the post-auricle and pre-pharynx level induces the formation of the unpigmented blastema, eye regeneration, and head remodelling (**a**,**b**). (**b**) This shows the regeneration of the anterior pole and the eyespots in a representative planaria treated with PAM alone 8 days post amputation. (**c**,**d**) This demonstrates a noticeable absence of anterior pole regeneration and development of eyespots. Representative images were taken at day 8 post amputation.

**Figure 3 pharmaceutics-15-02018-f003:**
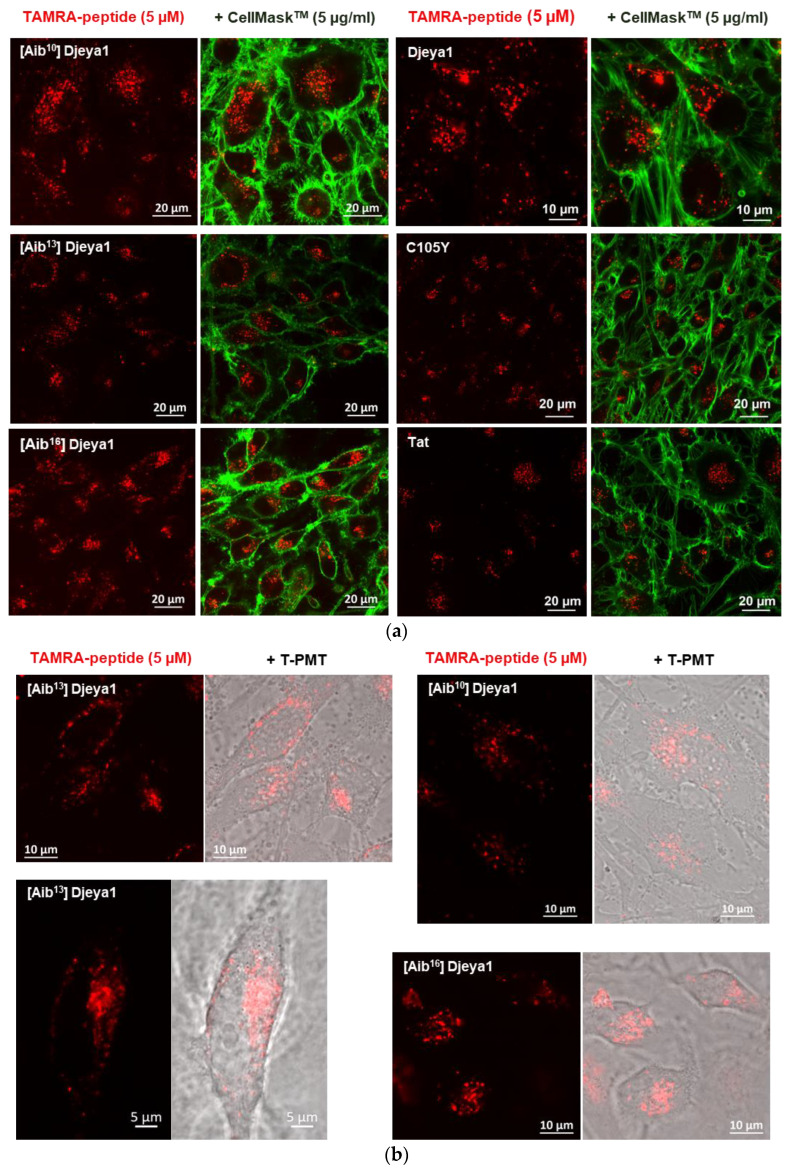
Qualitative uptake analyses using live confocal cell imaging. (**a**) U373MG cells were treated with 5 μM TAMRA-labelled peptides and 5 μg mL^−1^ CellMask^TM^ to label the plasma membrane for 1 h prior to visualization. Tat and C105Y were used as positive controls. (**b**) U373MG cells were treated for 5 h with 5 μM TAMRA-labelled Aib-substituted Djeya1 analogues. Confocal images are also presented with images using photomultiplier for transmitted light (T-PMT) to highlight subcellular distribution.

**Figure 4 pharmaceutics-15-02018-f004:**
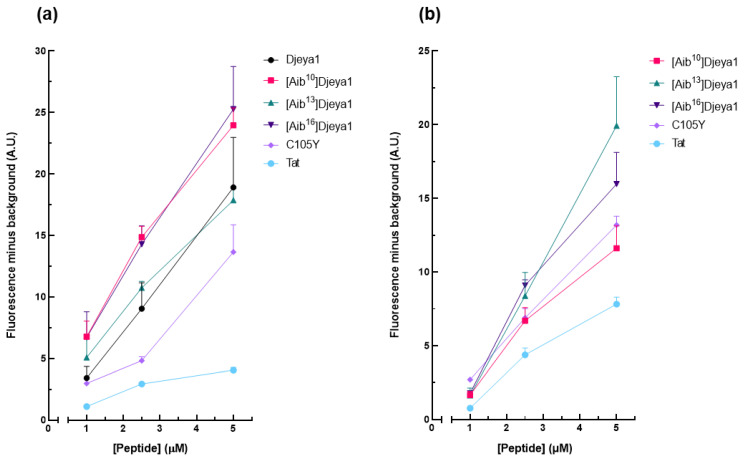
Quantitative peptide uptake analyses. Comparative analyses of peptide translocation efficacies of Aib-substituted Djeya1 analogues were performed using U373MG cells (**a**) and U251 cells (**b**) incubated with TAMRA-labelled peptides for 1 h at 37 °C at the concentrations indicated. Both Tat and C105Y were used as positive controls. Data are expressed as mean fluorescence minus background ± SEM. Data are from two experiments performed in triplicate.

**Figure 5 pharmaceutics-15-02018-f005:**
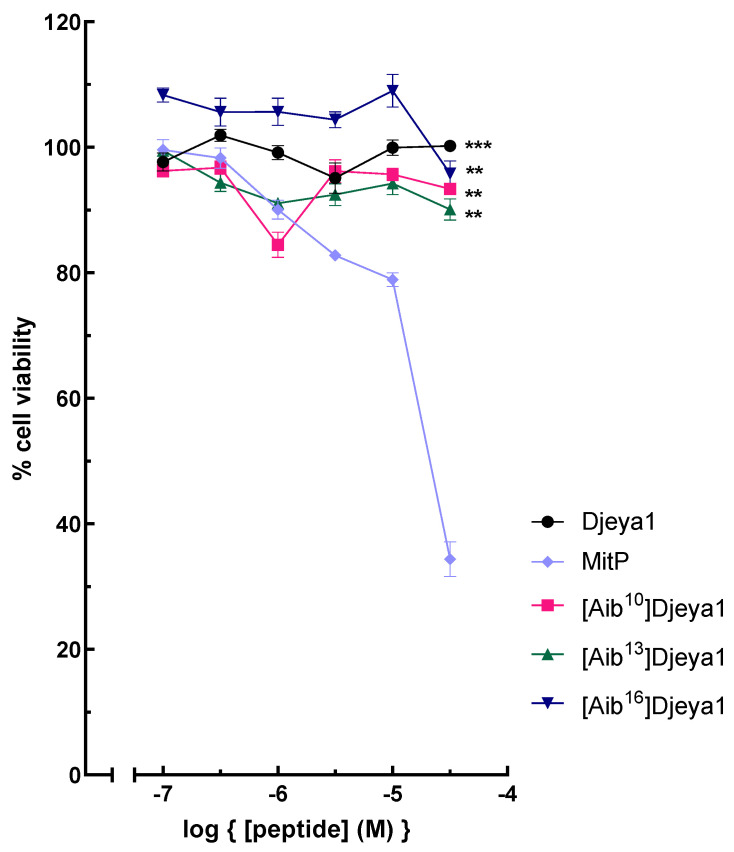
Cytotoxicity profiles of Aib-substituted Djeya1 analogues. U373MG cells were treated with Djeya1 and Aib-substituted analogues for 4 h at the concentrations indicated. Cell viability was measured by MTT conversion and expressed as a percentage of those cells treated with vehicle (medium) alone. The mitochondriotoxic peptide Mitoparan (MitP) was used as a positive control [11,31]. Data points are mean ± SEM from two experiments performed in triplicate. Statistical analyses employed a non-parametric Mann–Whitney test to compare changes in viability to that of MitP at 30 μM, (*** *p* = 0.0001, ** *p* = 0.0022), GraphPad Prism 9 software.

**Figure 6 pharmaceutics-15-02018-f006:**
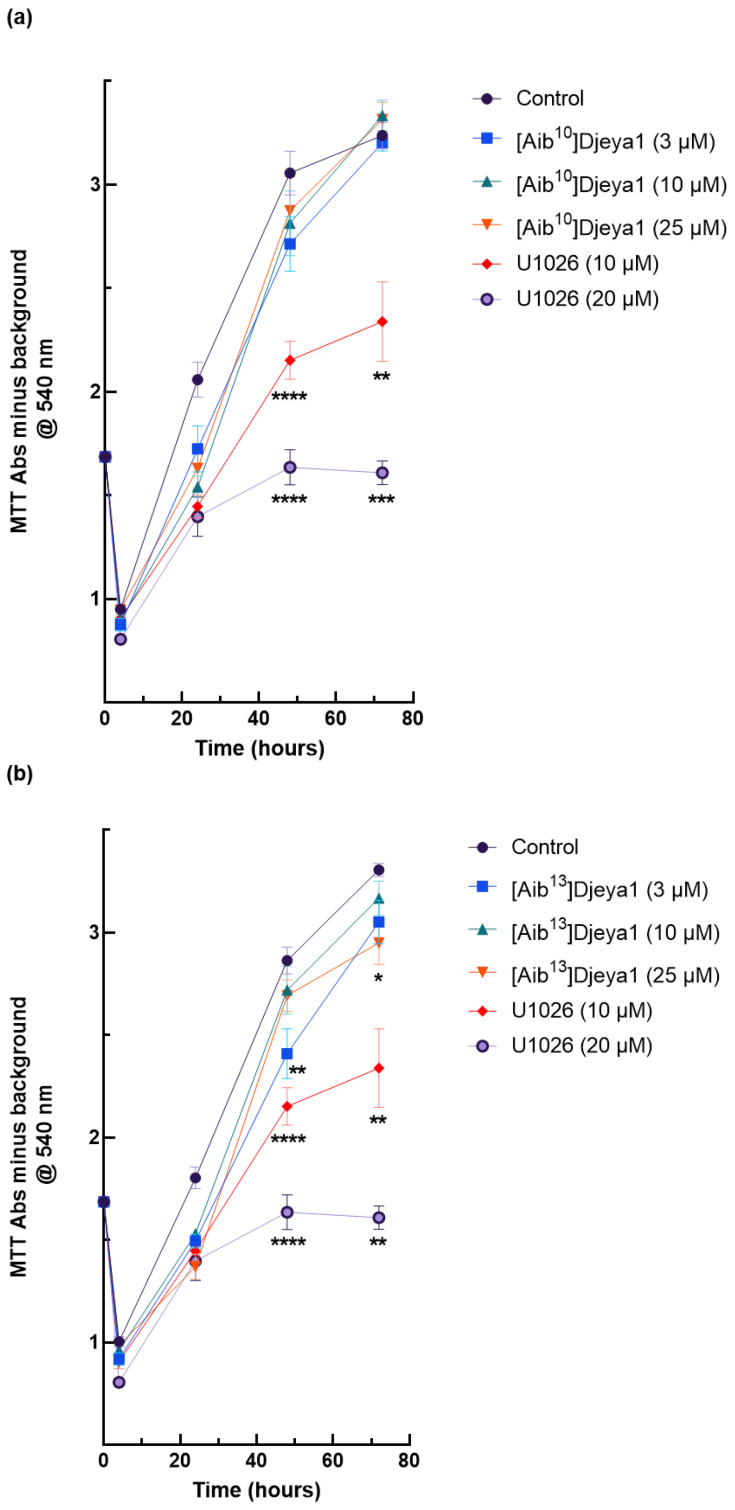
Growth curves of high Eya expressing U251 cells expressed as optical density. U251 cells were treated with Aib-substituted analogues or the p42/44 MAPK inhibitor U1026 at the concentrations indicated from 4–72 h (**a**–**c**). MTT conversion is expressed as Abs@540 nm minus background, and data points are mean ± SEM from two experiments performed in sextuplicate. A total of 4 h and 72 h, with one experiment performed in sextuplicate. Control denotes cells treated with medium alone. Statistical analyses comparing significant differences in cell viability to the untreated control were performed at 48 h and 72 h using the unpaired, 2-tailed, non-parametric Mann–Whitney test, (* *p* < 0.05, ** *p* < 0.005, *** *p* = 0.0001, **** *p* < 0.0001), using GraphPad Prism 9 software.

**Figure 7 pharmaceutics-15-02018-f007:**
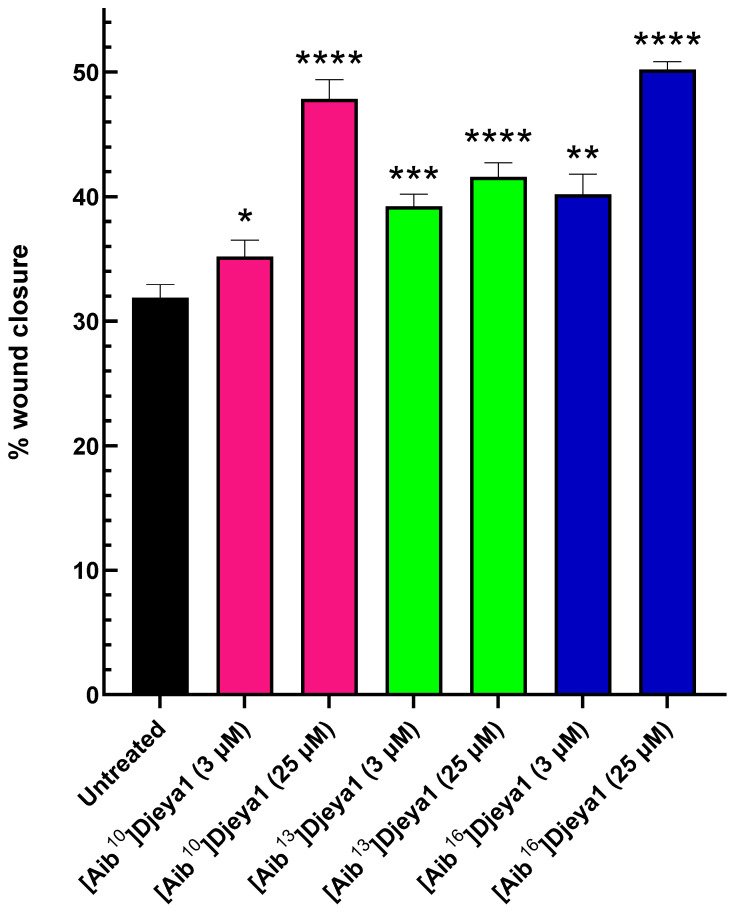
Aib-substituted Djeya1 analogues enhance PC-3 cell migration. Wound closure was measured after 48 h. All analogues demonstrated a significant enhancement in percentage wound closure compared to untreated cells, *p* < 0.05 using the paired, 2-tailed non-parametric Wilcoxon matched pairs signed rank test (* *p* = 0.0353, ** *p* = 0.0015, *** *p* = 0.0002, **** *p* < 0.0001), GraphPad Prism 9 software.

**Table 1 pharmaceutics-15-02018-t001:** Sequence conservation within the α5 helix of ED.

Species	Sequence
*Dugesia japonica (Djeya 1)* *Lingula unguis* *Euprymna scolopes* *Euperipatoides kanangrensis* *Dryobates pubescens* *Helobdella robusta* *Branchiostoma floridae* *Branchiostoma lanceolatum* *Capitella teleta* *Melanaphis sacchari* *Schizaphis graminum* *Felis catus* *Canus familiarus* *Bos Taurus* *Homo sapiens* *Ornithodoros moubata*	RKLAFRYRRIKELYNSYR RKLAFRYRRIKEIYNSYR RKLAFRYRRIKEIYNSYR RKLAFRYRRIKEIYNSYR RKLAFRYRRVKELYNTYR RKLAFRYRRIKELYSAYR RKLAFRYRRIKEIYNSYK RKLAFRYRRIKEIYNSYK RKLAFRYRRIKEIYSSYR RKLAFRYRRVKELYNQYR RKLAFRYRRVKELYNQYR RKLAFRYRRVKELYNTYK RKLAFRYRRVKELYNTYK RKLAFRYRRVKELYNTYK RKLAFRYRRVKELYNTYK RKLAFRYRRIKEIYNQYR

Consensus—RKLAFRYRR(I/V)KE(L/I)YN(S/T/A/Q)Y(R/K).

**Table 2 pharmaceutics-15-02018-t002:** Primary sequences of CPPs and bioportides.

Peptide	Sequence
Djeya [Aib^10^]Djeya1 [Aib^13^]Djeya1 [Aib^16^]Djeya1 Tat C105Y Mitoparan	*H*-RKLAFRYRRIKELYNSYR-*NH_2_* *H*-RKLAFRYRR(Aib)KELYNSYR-*NH_2_* *H*-RKLAFRYRRIKE(Aib)YNSYR-*NH_2_* *H*-RKLAFRYRRIKELYN(Aib)YR-*NH_2_* *H*-GRKKRRQRRRPPQ-*NH_2_* *H*-CSIPPEVKFNKPFVYLI-*NH_2_* *H*-INLKKLAKL(Aib)KKIL-*NH_2_*

**Table 3 pharmaceutics-15-02018-t003:** (a) Calculated secondary structures of Djeya1 analogues in PBS. (b) Calculated secondary structures of Djeya1 analogues in 50% (*v*/*v*) TFE. (c) Calculated secondary structures of Djeya1 analogues in DMPC vesicles. (d) Calculated secondary structures of Djeya1 analogues in DMPS vesicles.

**(a)**				
**Peptide**	**α-helix**	**ß-Strand**	**ß-Turns**	**Unordered**
Djeya1	43.00 ± 1.73	30.33 ± 0.58	6.67 ± 0.58	20.00 ± 1.73
[Aib^10^]Djeya1	41.33 ± 1.15	31.33 ± 0.58	7.00 ± 0.00	20.67 ± 0.58
[Aib^13^]Djeya1	40.67 ± 1.52	30.00 ± 1.00	7.00 ± 0.00	22.33 ± 1.52
[Aib^16^]Djeya1	39.67 ± 4.04	31.67 ± 2.87	7.33 ± 0.58	21.00 ± 1.00
**(b)**				
**Peptide**	**α-helix**	**ß-Strand**	**ß-Turns**	**Unordered**
Djeya1	45.50 ± 4.94	27.55 ± 3.53	7.00 ± 0.00	20.00 ± 1.41
[Aib^10^]Djeya1	51.50 ± 0.07	21.00 ± 0.02	7.50 ± 0.71	20.00 ± 0.00
[Aib^13^]Djeya1	40.00 ± 2.64	31.00 ± 2.08	7.33 ± 0.05	22.33 ± 1.50
[Aib^16^]Djeya1	46.00 ± 1.41	27.50 ± 0.71	7.00 ± 0.00	19.50 ± 0.71
**(c)**				
**Peptide**	**α-helix**	**ß-Strand**	**ß-Turns**	**Unordered**
Djeya1	44.00 ± 1.53	34.00 ± 2.12	8.00 ± 0.0	22.00 ± 0.00
[Aib^10^]Djeya1	45.00 ± 1.58	35.00 ± 0.70	7.00 ± 1.41	22.00 ± 1.41
[Aib^13^]Djeya1	42.00 ± 0.71	30.00 ± 0.71	8.00 ± 0.71	22.00 ± 0.71
[Aib^16^]Djeya1	42.00 ± 0.71	30.00 ± 0.71	7.00 ± 0.71	22.00 ± 0.71
**(d)**				
**Peptide**	**α-helix**	**ß-Strand**	**ß-Turns**	**Unordered**
Djeya1	47.00 ± 1.24	26.00 ± 1.28	8.00 ± 0.00	20.00 ± 0.00
[Aib^10^]Djeya1	63.00 ± 2.82	15.00 ± 0.00	5.00 ± 0.00	16.00 ± 0.00
[Aib^13^]Djeya1	42.00 ± 0.00	30.00 ± 0.71	7.00 ± 0.71	21.00 ± 0.71
[Aib^16^]Djeya1	42.00 ± 0.00	30.00 ± 0.71	8.00 ± 0.71	22.00 ± 0.71

## Data Availability

The data presented in this study are available on request from the corresponding author.

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
