# Peer review of "Stem Cell Bioengineering with Bioportides: Inhibition of Planarian Head Regeneration with Peptide Mimetics of Eyes Absent Proteins"

_pharmaceutics, 2023, doi:10.3390/pharmaceutics15082018_

Round 1

Reviewer 1 Report

In their paper, the authors present interesting findings on the inhibitory activity of Djeya1 peptides on the ability of neoblasts to regenerate the anterior pole in decapitated S. mediterranea-model organism used to study the regenerative capacity of stem cells and tissue development. I find these results interesting and valuable. The authors correlate the ability to adopt the helical conformation of the peptide with the efficiency of translocation across the cell membrane and their inhibitory properties. In order to increase the helicity of the peptides, the authors used the standard way of modifying them involving the introduction of an aminoisobutyric acid (Aib) residue into the structure of the peptide. I have no comments on the description of the peptide synthesis and cellular studies. The only minor comment is the lack of MS data for TAMRA-labeled peptides. For Aib-labeled peptides these data are provided.

My biggest concern is with the section on conformational analysis of peptides based on circular dichroism (CD).

The CD spectra presented are of poor graphical quality. If the publication is online, color CD curves could be introduced. The MRE scale for fig 1a is far too wide, which flattens the CD curves and makes them poorly distinguishable, especially those marked with dashed lines. Why do the theta values in fig 1b have such large values. For fully helical proteins, this value is below -40,000. In fig 1b, the values are -60 000 and even -80,000. Why?

The authors use a 50% TFE solution to simulate the cell membrane environment. TFE is not the best cell membrane mimetic. Much better are liposomes of phospholipids though such as DPC. Why didn't the authors use this type of compound to simulate the cell membrane environment? If TFE was already used, why in such a high concentration-50%. Usually a 30% solution is used in the literature. There should be an explanation of these problems in the text.

At the beginning of Section 3.2, the authors write that CD spectra containing extrema at 207 and 222 nm are characteristic of the alpha helical  and random coil (RC) conformation. For the alpha helical conformation, yes, but RC is characterized by a minimum at about 195 nm. Considering the intensity of these signals, one can indeed conclude that they show a mixture of the mentioned structures. Is this what the authors had in mind? I also noticed that the authors use the term Tat. In this form it is reserved for the full Tat protein. The peptide the authors use is Tat(48-60)-NH2. I would suggest at least changing the term Tat to Tat peptide. Having corrected these glitches, I recommend the manuscript for publication.

Author Response

xcv+X/C/V

Dear Reviewer,

Thank you for your suggestions which will no doubt improve the quality of this manuscript.

Mass spectrometry data for all three novel Aib-substituted Djeya 1 analogues have now been included in the supplementary section. TAMRA-conjugated peptides were produced using a very well-established method for the on-resin N-terminal acylation of peptides already characterised by mass spectrometry. 

CD Spectra

The referee suggested that the scaling on Figure 1 should be revised.  In response, the scaling was checked and a scaling error in figure 1b was revealed.  This has been corrected and new figures have been incorporated into the revised paper. We would like to thank the reviewer for bringing this to our attention.

“The authors use a 50% TFE solution to simulate the cell membrane environment. TFE is not the best cell membrane mimetic. Much better are liposomes of lipids though such as DPC. Why didn't the authors use this type of compound to simulate the cell membrane environment? If TFE was already used, why in such a high concentration-50%. Usually, a 30% solution is used in the literature. There should be an explanation of these problems in the text.”

It is general practice to use 50% TFE to mimic the anisotropic environment of cell membranes when studying the conformational behaviour of peptides (e.g. Gong et al 1). However, we accept the reviewer’s comment that liposomes formed from phospholipids would provide more appropriate and accurate mimics of these membranes.  Therefore, additional work has been undertaken using liposomes formed from DMPC and DMPS to mimic cell membranes.  These data have been incorporated into the text of the revised paper and the relevant sections updated.

[1] Gong, Z., Ikonomova, S. P., and Karlsson, A. J. (2018) Secondary structure of cell penetrating peptides during interaction with fungal cells, Protein Sci 27, 702-713

“At the beginning of Section 3.2, the authors write that CD spectra containing extrema at 207 and 222 nm are characteristic of the alpha helical and random coil (RC) conformation. For the alpha helical conformation, yes, but RC is characterized by a minimum at about 195 nm. Considering the intensity of these signals, one can indeed conclude that they show a mixture of the mentioned structures. Is this what the authors had in mind?”

We agree with the reviewer’s comment and have rephrased the first paragraph as has been suggested

So as to prevent any confusion between the Tat protein and Tat48-60, an explanatory component has been included in the legend to table 2.

Reviewer 2 Report

The manuscript entitled “Stem Cell Bioengineering with Bioportides: Inhibition of Planarian Head Regeneration with Peptide Mimetics of Eyes Absent Proteins” describes engineered analogs of cell-penetrating peptide Djeya1 – the mimic of a particular helix of the highly conserved domain of eyes absent proteins – and its improved properties. The molecular design was rather obvious and based on substitution of certain amino acids (namely, at positions 10, 13 and 16) within the sequence of a parent peptide against Aib. In general, substitution with α,α-amino acids, in particular with Aib, is not new and known since decades as a tool to increase helicity and cellular uptake of peptidic molecules. The authors state in the Introduction that “the major objective of this study was to determine whether the bioengineering of Djeya1 analogues could provide rhegnylogically organised bioportides, CPPs in which the pharmacophores that enable cellular penetration and those essential for bioactivity are discontinuously organized”. To that end, I would recommend to add more information on the issue of rhegnylogic and sychnologic art of bioportides organization and also show, which fragments in Djeya1 peptide are supposed to enable cell penetration, and which ones are responsible for bioactivity.

Taken that I am not an expert in planaria biology, I definitely like the manuscript. However, certain issues are not clear to me. The first one dealing with design I have already mentioned. The second one is the production of peptides. Considering that assembly of peptides containing Aib can be rather challenging, I understand that the authors would like to publish the synthesis of bioportides elsewhere. However, I would like to see at least an exemplarily MALDI or HR-ESI spectrum in a Supplementary Section. The next issue is the uptake of engineered bioportides in U373MG and U251 cells. It would be nice to give a small discussion on the possible mechanisms of uptake and comment on the reasons, why the variants have different uptake efficiency. Was the cellular uptake only analyzed at 37 °C?

I would also recommend to read the manuscript carefully for typos. For example, “Rink Amide 4-Methylbenzhydrylaamine (MBHA) resin - should be “methylbenzhydrylamine”, “AA/DIC/Oyxma” – should be “Oxyma”, and so on. Also the usage of capitals in the names of chemical compounds should be checked.

To summarize: in my opinion, this is not a groundbreaking research as it lacks novelty. Nevertheless, it is properly designed, sound, and clearly written. It can be interesting for the researches working in the field of CPPs and bioportides. I recommend its publication after all the mentioned concerns have been addressed.

Quality of English is satisfactory, but the manuscript should be checked for typos.

Author Response

Dear Reviewer,

Thank you for your suggestions which will no doubt improve the quality of this manuscript.

The definitions of sychnologic and rhegnylogic organisations of bioportides have now been addressed with reference to pharmacophores for cellular penetration and bioactivity.

From line 64:

Hence, the major objective of this study was to determine whether the bioengineering of Djeya1 analogues could provide rhegnylogically organised bioportides, CPPs in which the pharmacophores that enable cellular penetration and those essential for bioactivity are discontinuously organised” This is in contrast to a sychnologic organisation in which the pharmacophores for penetration and bioactivity are distinct. Moreover, given that Arg is the quantitatively dominant amino acid at PPI interfaces, there will clearly be some pharmacophores which are both cell penetrant and bioactive within a rhegnylogic organisation [4,5].

Spectra have now been included in the Supplementary Section and referred to in the legend of Table 2.   

“The next issue is the uptake of engineered bioportides in U373MG and U251 cells. It would be nice to give a small discussion on the possible mechanisms of uptake and comment on the reasons, why the variants have different uptake efficiency. Was the cellular uptake only analyzed at 37 °C?”

The discussion has been expanded to address these points.

The bioactivities of [Aib13]Djeya1 and [Aib16]Djeya1 cannot be readily explained by increased uptake into cells. Both qualitative and quantitative analyses show that the uptake of all Aib-substituted Djeya1 analogues are generally comparable to that of the parent CPP and better than other CPPs including Tat and C105Y. It is however notable that a different rank order of penetrative efficacy was evident between U373MG and U251 cell lines, which in part may be attributable to the U251 cell line overexpressing the EYA protein [29]. We have previously proposed [22] that the propensity for cellular penetration and intracellular accumulation of CPPs involves 2 distinct processes: 1, translocation across the plasma membrane (hydrophobicity and cationic charge are significant factors); 2, accretion at intracellular loci. Thus, the increased uptake of [Aib13]Djeya1 within U251 cells, could reflect in part enhanced accretion at intracellular loci, particularly since [Aib10] Djeya1, also with a conservative aliphatic side chain substitution for Aib, showed a reduced uptake efficacy compared to [Aib13]Djeya1.

It would be intriguing to establish the mechanism of uptake for these novel Aib-substituted Djeya1 analogues. Following live confocal cell imaging, all analogues showed a clear punctate intracellular distribution, data which suggest an endocytotic mechanism of intracellular uptake. Further investigations, including the co-labelling with specific markers of endocytosis or micropinocytosis, in addition to quantitative uptake analyses performed at both 4 °C and 37 °C, could define a mechanism of cellular uptake. Additionally, all analogues demonstrated a distinct absence of cytotoxicity even at a peptide concentration of 30 μM. Thus, these peptides do not induce gross perturbations of the plasma membrane to induce Ca2+-dependent necrosis.

Typos in the Methods Section have now been addressed.

“To summarize: in my opinion, this is not a groundbreaking research as it lacks novelty”

Thank you for making this point as a clearer explanation of the novel aspects of this study warrant highlighting. The novelty of this submission is not our established bioportide technologies alone, but rather their use to modulate planarian stem cell biology by targeting PPIs integral to anterior pole and eye regeneration. Thus far, such investigations have predominantly used siRNA interference and to our knowledge, this is the first instance in which bioportide technologies have given a functional response in this three-dimensional model. Logically, this is a natural progression from our previous findings which established S.mediterranea as a model for the discovery and characterization of CPPs and bioportides (Jones et al., 2019)

Jones, S.; Osman, S.; Howl, J. The planarian Schmidtea mediterranea as a model system for the discovery and characterization of cell penetrating peptides and bioportides. Chem. Biol. Drug Des. 2019, 93, 1036-1049

This has been emphasized in the conclusion.

Reviewer 3 Report

In this manuscript, the authors bioengineered analogues of Djeya1 to reduce the ability of neoblasts (totipotent stem cells) and their progeny to regenerate the anterior pole in decapitated S. mediterranea after efficient transfer into planarian tissues. The results indicated that Aib had no effect on cell viability or proliferation and increased the migration of PC-3 cells. Overall, this topic is fascinating and provides novel information. However, the manuscript lacked logical coherence and the discussion was not sufficiently detailed. Therefore, I highly recommend accepting the manuscript with minor revisions.

  Some specific issues should be addressed as follows:

1.        The uniformity of all images and tables needs to be enhanced, including Figure 1, Table 3, and so on.

2.        The detailed methods for statistical analysis should be described in the Methods section.

3.        The 2D structure of Djeya1 should be provided. Moreover, the interaction between membranes and peptides should provide more conclusive evidence, in addition to CD analysis.

4.        In section 3.4.2, the authors should have strengthened the discussion of the quantitative data.

5.        In section 3.7, what is the mechanism by which the Djeya analogue replaced by Aib promotes cell migration? Please provide an explanation.

6.        A conclusion section should be added to summarize the findings of the manuscript.

7.   The author must thoroughly check the manuscript for any other mistakes. 

The manuscript should be carefully checked to avoid the spelling, expression and grammar errors. 

Author Response

The authors would like to thank the reviewer for their constructive comments which will no doubt enhance this manuscript.

  1. The uniformity of all images and tables needs to be enhanced, including Figure 1, Table 3, and so on.

Figure 1 and Table 3 have been enhanced owing to requested further experiments and improved for uniformity.

  1. The detailed methods for statistical analysis should be describedin the Methods section.

This has now been included.

  1. The 2D structure of Djeya1 should be provided. Moreover, the interaction between membranes and peptides should provide more conclusive evidence, in addition to CD analysis.

These analyses compare Djeya1 with three bioengineered analogues. This section has been expanded to include additional experiments in the presence of both anionic and zwitterionic lipid vesicles.

  1. In section 3.4.2, the authors should have strengthened the discussion of the quantitative data.

The discussion regarding quantitative uptake analyses using U373MG and U251 cell lines has now been expanded.

  1. In section 3.7, what is the mechanism by which the Djeya analogue replaced by Aib promotes cell migration? Please provide an explanation.

As we state in the manuscript, these data were unexpected. The rationale for these experiments was to check whether those analogues which inhibit head regeneration would also inhibit cellular migration, thus suggesting that bioportides prevent neoblast migration to the site of wounding. Clearly, this was not the case. Moreover, given that [Aib10]Djeya 1 does not inhibit head morphogenesis, it is unlikely that this positive influence on cell migration underlies the mechanism of action of [Aib13]Djeya 1 and [Aib16]Djeya 1 to delay anterior pole regeneration. To determine a mechanism by which all three peptides promote cellular migration would require many additional investigations that are simply beyond the scope of this investigation and would not change the key conclusions of the study.

  1. A conclusion section should be added to summarize the findings of the manuscript.

The manuscript now has a conclusions section.